# The Predictive Power of the 14–51 Ng/L High Sensitive Troponin T (hsTnT) Values for Predicting Cardiac Revascularization in a Clinical Setting

**DOI:** 10.3390/jcm11237147

**Published:** 2022-12-01

**Authors:** Frank W. De Jongh, Sjaak Pouwels, Marjolein C. De Jongh, Eric A. Dubois, Ron H. N. van Schaik

**Affiliations:** 1Department of Clinical Chemistry, Erasmus University Medical Center, 3015 GD Rotterdam, The Netherlands; 2Department of Cardiology, Erasmus University Medical Center, 3015 GD Rotterdam, The Netherlands; 3Department of Cardiothoracic Surgery, HAGA Hospital, 2545 AA The Hague, The Netherlands; 4Department of Intensive Care Medicine, Elisabeth-Tweesteden Hospital, 5011 GB Tilburg, The Netherlands; 5Department of Cardiology, HAGA Hospital, 2545 AA The Hague, The Netherlands

**Keywords:** high sensitive troponin T, hsTnT, cardiac revascularization, myocardial infarction, ACS, non-STEMI

## Abstract

Background: high sensitive Troponin T (hsTnT) values between 14–50 ng/L represent a challenge in diagnosing acute coronary syndrome (ACS) at the Emergency Department (ED). The European Society for Cardiology (ESC) recommends a second hsTnT measurement 3 h later to distinguish between ACS and other causes depending on the Δ hsTnT. Our study aims to evaluate the predictive power this approach in a clinical setting by following patients presenting at the ED with hsTnT values 14–51 ng/L. Materials and methods: patients presenting with chest pain or dyspnea and a hsTnT value between 14 and 50 ng/L at the Erasmus MC ED in 2012–2013 were included and retrospectively monitored for 90 days after initial presentation for the occurrence of a cardiac revascularization. Patient records were reviewed according to the standing protocol, which depended on the Δ hsTnT. The “event-group” consists of patients receiving cardiac revascularization within 90 days after the ED visit, whereas the “no event-group” consisted of patients without revascularization. Results: a total of 889 patients patient records were reviewed. After excluding out-of-hospital-cardia-arrests (60), non-cardiological chest pain (373) and incomplete follow-up (100), 356 patients remained for final analysis. In 207 patients, a second hsTnT was actually performed (58%). From these 207 patients, 68 (33%) had a Δ hsTnT ≥7 ng/L. In these patients, 37 (54%) experienced an event within 90 days. In the 139 patients with a Δ hsTnT < 7 ng/L, 23 (17%) presented with an event within 90 days. Conclusion: our study demonstrated a sensitivity of 62%, a specificity of 79%, a positive predicted value (PPV) of 54% and a negative predictive value (NPV) of 83% for using a 3-h Δ hsTnT ≥7 ng/L cut-off, related to risk of an event in 90 days following ED presentation.

## 1. Introduction

In patients presenting with chest pain with a non-ST-segment elevation myocardial infarction (NSTEMI) or a ST-elevated myocardial infarction (STEMI), increased levels of troponin due to cardiomyocyte injury are found [1,2,3]. In these patients, levels of cardiac troponin rise rapidly after symptom onset and remain elevated for a variable period of time (usually several days) [1,2,3]. Other conditions that can be associated with elevated levels of cardiac troponin are tachyarrhythmias, heart failure, hypertensive emergencies, aortic dissection, renal dysfunction, and many other diseases [1,2,3].

To identify patients having a myocardial infarction (MI), algorithms have been developed to enhance the diagnostic pathway for patients presenting at the Emergency Department (ED) with chest pain. Cut-off levels of troponin are assay-specific [4,5,6,7,8]. The algorithm (Figure 1) used in the Erasmus Medical Center Rotterdam (EMC) is based on the publication of Reichlin et al [5] and the European Society of Cardiology (ESC) guidelines for use of Troponin for NSTEMI. In the absence of STEMI, a patient is considered to have Acute Coronary Syndrome (ACS) when hsTnT > 50 ng/L at presentation. Patients with a hsTnT < 14 ng/L with >3 h of pain are deemed not to be at risk for a MI. Patients presenting with chest pain and hsTnT 14–50 ng/L cannot be diagnosed on biomarkers and clinical presentation alone., therefore a second hsTnT is determined 3 h after the first measurement. In case of a Δ hsTnT ≥7 ng/L, the diagnosis NSTEMI is considered likely. The positive predictive value for Myocardial Infarction (MI) in those patients was reported to be 75–80% [6,7,8]. 

Values of 14–50 ng/L represent a group of patients with intermediate risk of acute cardiac ischemia. When the initial hsTnT value is 14–50 ng/L, a Δ hsTnT < 7 ng/L after 3 h indicates that ACS is unlikely, and the patient can be either discharged or treated as unstable angina pectoris. According to the ESC guidelines, this heterogeneous group requires further investigations if no alternative explanation for the cardiac troponin elevation has been identified [1,2]. If there is a clinical high suspicion of NSTEMI, coronary angiography should be considered. In patients with a lower to intermediate suspicion, computed tomography (CT) coronary angiography may be considered. If there is another alternative diagnosis, such as tachyarrhythmias, high blood pressure, or renal dysfunction, no further diagnostic testing is recommended [1,2].

We aimed to retrospectively assess the existing protocol and find the predictive value of the Δ hsTnT ≥7 ng/L 3 h after the first assessment protocol in the patient group presenting at the ED with 14–50 ng/L in a real-life clinical setting by monitoring patient follow-up for the occurrence of cardiac revascularization for 90 days after the first ED visit.

## 2. Materials and Methods

All hsTnT level lab results between 14 and 51 ng/L measured at the ED of the Erasmus MC in the periods of 2012 and 2013 were identified. Second hsTnT values were collected for these patients when available. Patients were excluded if they presented with complaints not suspected of cardiac origin (based on clinical presentation, ECG, and biomarkers) (like stomach ache), out-of-hospital-cardiac-arrest (OHCA), clinically evident STEMI, or when lost to follow-up. The remaining patients were categorized on whether or not they received cardiac revascularization within 90 days of their ED presentation (event/no event groups). Baseline characteristics were recorded, as well as the time between the start of complaints and the first hsTnT and second hsTnT measurements. The difference in the first and second hsTnT values was used to categorize patients with a Δ hsTnT ≥7 or <7 ng/L [1,2]. Table 1 gives an overview of all hsTnT measurements. Revascularization rates, including percutaneous coronary intervention (PCI) and coronary arterial bypass grafting (CABG) within 90 days, were evaluated.

### 2.1. Study Population

Patient characteristics of interest were gender, age, reasons for ED visits (defined as chest pain, dyspnea or other), and cardiac history (defined as none, coronary artery bypass graft (CABG), percutaneous coronary intervention (PCI), infarction without revascularization, other heart decease or unknown).

### 2.2. Sample Collection and Laboratory Testing

Peripheral venous blood samples were taken after initial presentation at the Emergency Department. hs-TnT was determined in thaw serum. Erasmus protocol states a result after 45 minutes of blood draw. No specific storage protocol is in place. 

All analyses were performed in the central chemistry laboratory of the Erasmus MC. We used Roche electrochemiluminesence immunoassays (Roche Diagnostics, Basel, Switzerland). Lower limits of detection were 5 ng/L for hs-TnT. The upper limit of normal was defined as the 99th percentile of the reference distribution, which corresponded with 14 ng/L for hs-TnT.

### 2.3. Statistical Analysis

Data were analyzed retrospectively. Data management and analysis were performed using Statistical Package for Social Sciences (SPSS, Chicago, IL, USA Version 20.0). Continuous variables were presented as mean ± standard deviation (SD) and categorical variables as the frequency with percentages. Dependent on the distribution and type of variable, either student *t*-test, Mann Whitney U-test, Chi-square test, or Fisher’s exact test was used to determine any statistical significance between the observed differences among groups. Differences were considered significant in the case of a *p*-value <0.05.

The corresponding patients were found in the electronic patient database (EPD) of the Department of Cardiology of the Erasmus Medical Center. Patient follow-up was set to one year. No formal ethical approval was required for this study. The Institutional Review Board for this retrospective study granted permission. This study has been performed in accordance with the ethical standards as laid down in the 1964 Declaration of Helsinki and was conducted according to the Strengthening the Reporting of Observational Studies in Epidemiology (STROBE) statement.

## 3. Results

A total number of 1,214 hsTnT values between 14 and 51 ng/L were registered at the Erasmus ED during the study period. This resulted in 889 unique patients, who were subsequently included in this study and retrospectively analyzed. After excluding patients with OHCA (*n* = 60), those with no suspected cardiac origin of complaints after anamnesis by ED physician and/or cardiologist (*n* = 373) and those who were lost during follow-up (*n* = 100), a total of 356 patients remained for further analysis (Figure 2). In 207 patients, a second hsTnT was performed (58%). From these 207 patients, an event occurred in 60 cases (29%). Looking at a Δ hsTnT criteria, 68 (33%) had a Δ hsTnT ≥7 ng/L, indicating them at risk. In this group, 37/68 (54%) indeed experienced an event within 90 days. In the 139 patients with a Δ hsTnT < 7 ng/L, indicating low risk, 23 (17%) presented with an event within 90 days.

In 149 patients (42%), no second hsTnT was determined, although this would have been appropriate in case of only taking into account the hsTnT value. Of these, 15 patients experienced an event. Seven of these patients immediately received a heart catheterization (CAD) because of a high suspicion of cardiac pathology (five of them had a STEMI). Of the other eight, four patients were admitted and received a revascularization later (all of them within 8 days). Of the 4 patients that were sent home, one patient initially presented with a STEMI but was chosen not to treat. The other three received elective treatments within 90 days after presentation.

Most of the patients with a Δ ≥7ng/L were admitted at the Erasmus MC with either an ACS (but treated conservatively due to age or comorbidities) or were admitted with an alternative diagnosis, presenting with increased hsTnT levels (e.g., heart failure, pulmonary embolism, aortic dissection) (*n* = 26). Five patients with a Δ hsTnT ≥7ng/L were discharged, four were lost to follow-up and one died within one year of unknown cause. Of the patients in which a second troponin was not determined (*n* = 134), 19 patients (14%) were admitted to the Cardiology Department for further analysis, 45 patients (34%) were admitted to other departments (of whom 10 died within the same year of non-cardiac cause) and 34 patients (25%) were transferred to another hospital.

### 3.1. Discharged Patients with Revascularization < 90 Days

In total, 24 patients with a hsTnT 14–50 ng/L of the event group (*n* = 75: 60 with a second hsTnT measurement and 15 without a second hsTnT test) received a revascularization procedure within 90 days (whether they were sent home or initially admitted and later discharged). In total, 9 out of 24 patients (38%) that were sent home with a ∆ hsTnT < 7 ng/L returned to the hospital with cardiac pathology, implying that this was also present at initial presentation. 

### 3.2. Revascularization and Second hsTnT Measurements

Figure 3 gives an overview between the delta of the values of the second hsTnT measurement in the patients that had a cardiac revascularization and the patients that received conservative treatment. In the 60 patients that had a revascularization 37 of them had a second hsTnT with a rise of ≥7 ng/L (62%). Of the 147 patients treated conservatively (and no revascularization), only 31 had a ∆ hsTnT ≥7 ng/L (21%). This indicates that two consecutive hsTnT tests are 74% accurate in assessing whether or not a patient presenting with thoracic pain or dyspnea will undergo a cardiovascular revascularization in 90 days after presentation.

## 4. Discussion

In this retrospective study, we aimed to identify the value of a second hsTnT value in a real clinical setting for the detection of cardiac events. We showed that two consecutive hsTnT tests had a 74% accuracy in assessing whether or not a patient presenting with thoracic pain or dyspnea, requiring cardiovascular intervention 90 days after presentation at the Emergency Department. In the patients in which a second hsTnT was measured but did not show a ≥7 (*n* = 139), 17% (*n* = 23) still had revascularization within 90 days, giving a positive predictive value for myocardial infarction of patients presenting with chest pain or dyspnea and having a 2nd hsTnT ≥ Δ7 in our study is thus 54%, whereas the negative predictive value (NPV) was 83%. The negative value we experienced is thus substantially lower than that described in the literature. These values are in line with previous studies [6,7,8]. 

Body et al. investigated whether hsTnT could immediately exclude an acute MI by means of a novel ‘rule out’ cut-off, where patients could be ‘ruled out’ only if they have a troponin concentration below this threshold and no evidence of ECG ischemia at the time of ED presentation [6,7]. In their study conducted in 2011, they evaluated the value of hsTnT in acute myocardial infarction (AMI) at the ED. They found only 1 patient (0.6%) with initially undetectable hsTnT with subsequent elevation (to 17 ng/L) and therefore the authors conclude that undetectable hsTnT is able to rule out AMI in the majority of patients on the ED presenting with chest pains [7]. This is in contrast to our study, as we looked at cardiac revascularization, other than AMI.

In a study by Giannitis et al. [9], a total of 2525 patients who presented with suspected NSTE-ACS to the ED, 280 (11%) were diagnosed with unstable angina (UA), defined as unstable symptoms and either undetectable (<5 ng/L; *n* = 22), normal (5–14 ng/L, *n* = 156) or stable elevated hsTnT (14–50 ng/L, *n* = 102). Mortality rates at 12 months for these groups were 0% (hsTnT < 5 ng/L), 1.9% (hsTnT 5–14 ng/L), and 6.9%, (hsTnT 14–50 ng/L). In the stable elevated group (*n* = 102), 8 patients (7.8%) were discharged, and no follow-up data is given on this group. Patient age does not change the cut-off hsTnT value to rule in ACS, although it can introduce bias when not interpreted correctly. The 14 ng/L was first established from a reference population with ages between 20 and 70 years [10]. The majority of patients presenting on the ED are older with many comorbidities and elevated hsTnT levels were found in 22% of healthy subjects over 70 [11,12,13,14].

Visale & Jaffe reviewed the newer hsTnT assays and found that they might have a lower specificity for ACS as compared with the more conventional assays, this mostly concerns elderly patients, diabetics, and those who are post-operative, since they are at risk for ACS and can present with hsTnT or ECG abnormalities [15]. Although patients presenting on the ED are not necessarily representative of the population, this may introduce bias in diagnosing ACS [16,17].

## 5. Limitations

Our study has its limitations. It is a retrospective study, which limits the generalizability of our results. Secondly, our study addresses the use of hsTnT in a real-life clinical setting, therefore it is not always clear how exact the protocols were followed. The high number of patients not receiving a second hsTnT measurement, although being in the target window of 14–50 ng/L for their first measure, demonstrates this. Of our population 11% (100/889) patients were lost to follow-up due to various reasons, this could provide selection bias. Although it is impossible to know if patients underwent a cardiac event <90 days and were admitted to another hospital. Furthermore, 42% (149/356) patients did not undergo a second hsTnT blood draw, this is partly due to cardiac symptoms being so clear that a second hsTnT blood draw was not deemed necessary, or that protocol was not followed correctly. This will be a topic of further (prospective) study.

Lastly, the 2020 NICE guidelines discuss the relevance of sex for diagnosis and risk stratification in patients admitted to the ED with NSTEMI using the hsTnT protocol. Since this retrospective article focuses on the predictive power of hsTnT in general, these factors were not taken into account [18]. This will also be a topic of further (prospective) study.

## 6. Conclusion

Our study shows that the use of two consecutive hsTnT tests for patients presenting at the ED with complaints of chest pain and having a hsTnT in the 14–50 ng/L range results in 74% accuracy in predicting a cardiovascular event (and possible cardiac revascularization) within 90 days of presentation. Moreover, using a ∆ hsTnT ≥7ng/L cut-off for a second hsTnT measurement, 3 h after the first measurement, gives a 62% sensitivity and a 79% specificity for predicting a cardiovascular event. From the patients not presenting a ∆ hsTnT ≥7ng/L, it is important to realize that 21% still experienced a cardiovascular event within 90 days. Further research is warranted to substantiate our findings and to further optimize the diagnostic process with regard to biomarkers in acute myocardial infarction.

## Figures and Tables

**Figure 1 jcm-11-07147-f001:**
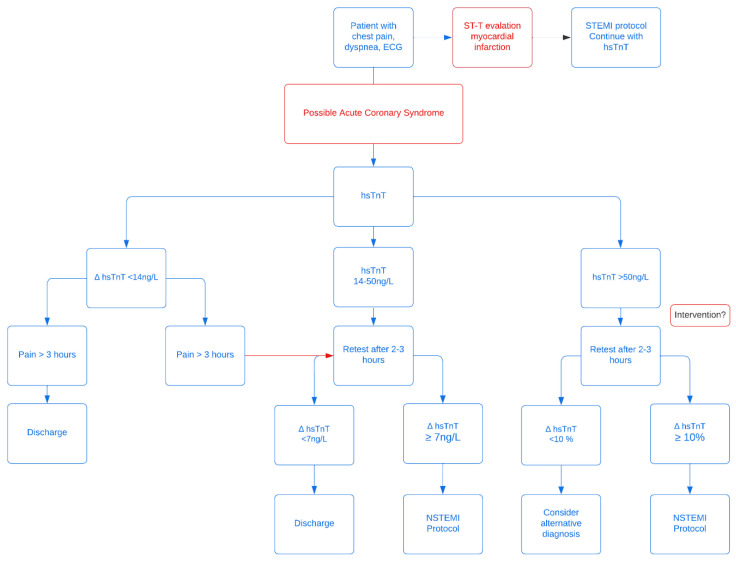
High sensitive Troponin T (hsTnT) protocol used in the Erasmus Medical Center.

**Figure 2 jcm-11-07147-f002:**
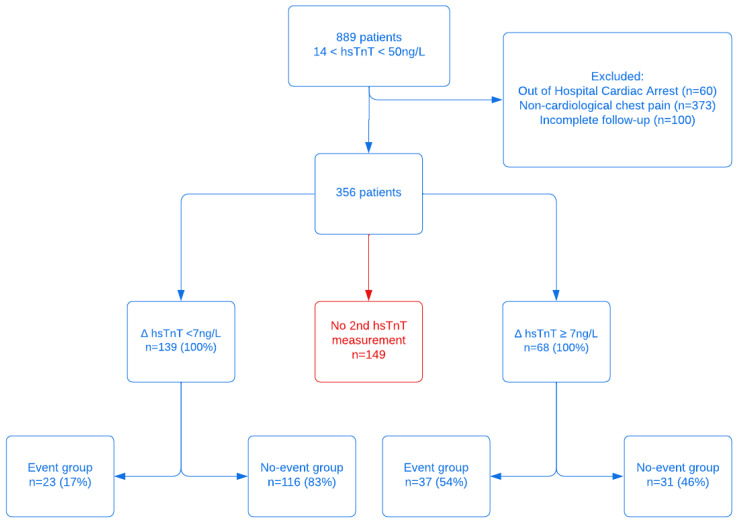
Patient inclusion flowchart. Event is defined as a revascularization within 90 days after initial presentation at the Emergency Department (ED) of the Erasmus MC Hospital.

**Figure 3 jcm-11-07147-f003:**
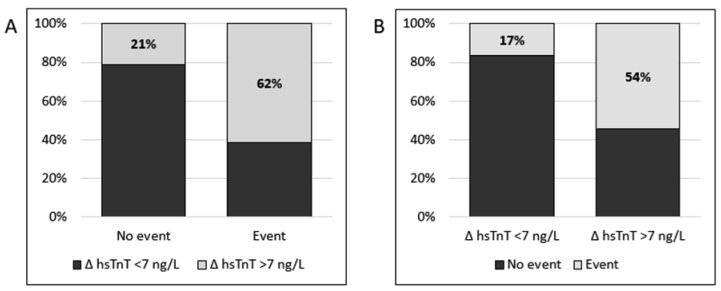
Overview of the delta (Δ) of the values of the second high sensitive Troponin T (hsTnT) measurements in the patients that had an event (*n* = 60) or did not have an event (*n* = 147) (**A**) or expressed as event/no event according to Δ hsTnT status at the ED (**B**).

**Table 1 jcm-11-07147-t001:** Characteristics of the patient in the event and the no-event group.

		Event < 90 Days	%	No Event < 90 Days	%
		*n* = 75		*n* = 281	
Sex	Male	60	80%	169	60%
	Female	15	20%	112	40%
Age (years) Complaints		63.7 ± 11.6		67.0 ± 13.5	
	Chest pain	71	95%	117	42%
	Dyspnoea	4	5%	120	43%
	Both	9	12%	44	12%
Cardiac History	None	27	36%	78	28%
	CABG	10	8%	17	6%
	PCI	23	31%	48	17%
	CABG + PCI	6	8%	16	6%
	Infarct without revascularization	2	1%	21	6%
	Other heart disease	6	2%	99	27%
	Unknown	1	0%	2	1%
Time of complaints until 1st hsTnT blood draw	<3 h	44	59%	53	19%
	3–6 h	5	7%	21	7%
	>6 h	20	27%	173	62%
	Unknown	6	8%	34	12%
Other parameters	1st hsTnT (average)	29.3 ± 10.3		26.3 ±10.0	
	Kidney Function eGFR (mL/min)	68.5 ± 18.4		59.6 ± 21.3	
	CK-MB (mass)	3.7 ± 2.9		3.5 ± 2.6	
	HsTnT drawn previously (>12 h)	20	27%	87	31%
Second hsTnT	2nd hsTnT (average)	440.02 ± 1130		53.51 ± 214.1	
	≥7	37	49%	31	11%
	<7	23	31%	116	41%
	Not drawn	15	20%	134	48%
Admissions	Dicharged	6	8%	114	41%
	Admitted/transferred	69	92%	159	57%
	Deceased	0	0%	8	3%
Cardiac Event	PCI	6	8%	x	x
	CABG	69	92%	x	x
Infarction		52	69%	11	4%

Abbreviations: CABG = Coronary Artery Bypass Grafting, PCI = Percutaneous Coronary Intervention, hsTnT = high sensitive Troponin T, eGFR = estimated Glomerular Filtration Rate, CK-MB = Creatinine Kinase-Muscle Brain.

## Data Availability

Data is available upon reasonable request.

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
