# Peer review of "The Predictive Power of the 14–51 Ng/L High Sensitive Troponin T (hsTnT) Values for Predicting Cardiac Revascularization in a Clinical Setting"

_jcm, 2022, doi:10.3390/jcm11237147_

Round 1

Reviewer 1 Report

To the Authors

General Considerations

The aim of this study was to to identify the value of a second hs-TnT value in a real clinical setting for the detection cardiac events by means of a retrospective study including a total number of 356 patients, but only 207 performed two hs-cTnT evaluations. Authors showed that two consecutive hsTnT tests had a 74% accuracy in assessing whether or not a patient presenting with thoracic pain or dyspnea, requiring cardiovascular intervention in 90 days after ED presentation. Te most important results of this study were that an algorithm using a 3 hour Δ hsTnT 7 ng/L cut-off, related to risk of an event in 90 days following ED presentation has a sensitivity of 62%, a specificity of 79%, a PPV of 54% and a NPV of 83% , corresponding to an efficiency of 74% and a LR of 2.9.

The article text is concise and the results reported are in part original. I would like to address to Authors some specific points with the aim to improve the scientific message of this interesting study.

Specific Points

1.     Materials and Methods. Some indications on the analytical characteristics and performance of the hs-cTnT method used in this study should be reported in details, in particular: the limit of detection (LOD), the 99th percentile value, and the analytical error in the range from 7 ng/l to 51 ng/L. If these analytical data are not available in the laboratory of the study, Authors should cite data from literature (as an example: Clerico A et al. Adv Clin Chem 2019;93:239-63). Furthermore, Authors should report some data on the collection (serum of plasma samples), the time of assay after the collection, and/or eventual storage of plasma/serum samples before assay. 

2.     Study population. Authors should report with more details in the text (Study Population section) the number of patients enrolled in the study and the number of patients divided in the different steps of enrollment and the experimental protocol, in accordance with the protocol summarized in the Figure 2.

3.     Statistical Analysis. I assume that the variables age and sex of the enrolled patients are not considered in the statistical analysis of this study. The relevance of age in the evaluation of cardiovascular risk using the delta changes of hs-cTnI and hs-TnI in the general population has been discussed in two recent authoritative documents (Farmakis et al. Eur Heart J 2020; 41: 4050-6; Clerico A et al. Clin Chem Lab Med 2021; 59: 79-90). Furthermore, the 2020 NICE guidelines (NICE. High-sensitivity troponin tests for the early rule out of NSTEMI. Diagnostics guidance. Published: 26 August 2020 www.nice.org.uk/guidance/ dg40) discussed in details the relevance of sex for diagnosis and risk stratification in patients admitted to ED with NSTEMI using the 0-3 hour algorithm. Authors should explain why the sex and age are not taken into consideration in this study. Authors should offer an explanation for this omission in the Statistical Analysis or (better) in the Limitations of the study. Of course, I believe that the scientific message of this study will be greatly improved if the analysis of sex and age as independent variables is taken into consideration in the revised version of the manuscript.

Author Response

Reviewer 1

  1. Materials and Methods. Some indications on the analytical characteristics and performance of the hs-cTnT method used in this study should be reported in details, in particular: the limit of detection (LOD), the 99thpercentile value, and the analytical error in the range from 7 ng/l to 51 ng/L. If these analytical data are not available in the laboratory of the study, Authors should cite data from literature (as an example: Clerico A et al. Adv Clin Chem 2019;93:239-63). Furthermore, Authors should report some data on the collection (serum of plasma samples), the time of assay after the collection, and/or eventual storage of plasma/serum samples before assay. 

Reply: An extra header is placed in the Methods section stating the time and collection method, sadly not all analytical data is available to us. This is now also mentioned.

  1. Study population. Authors should report with more details in the text (Study Population section) the number of patients enrolled in the study and the number of patients divided in the different steps of enrollment and the experimental protocol, in accordance with the protocol summarized in the Figure 2.

Reply: This data with details on the division in different steps is summarized in the results.

  1. Statistical Analysis. I assume that the variables age and sex of the enrolled patients are not considered in the statistical analysis of this study. The relevance of age in the evaluation of cardiovascular risk using the delta changes of hs-cTnI and hs-TnI in the general population has been discussed in two recent authoritative documents (Farmakis et al.Eur Heart J 2020; 41: 4050-6; Clerico A et al. Clin Chem Lab Med 2021; 59: 79-90). Furthermore, the 2020 NICE guidelines ( High-sensitivity troponin tests for the early rule out of NSTEMI. Diagnostics guidance. Published: 26 August 2020 www.nice.org.uk/guidance/ dg40) discussed in details the relevance of sex for diagnosis and risk stratification in patients admitted to ED with NSTEMI using the 0-3 hour algorithm. Authors should explain why the sex and age are not taken into consideration in this study. Authors should offer an explanation for this omission in the Statistical Analysis or (better) in the Limitations of the study. Of course, I believe that the scientific message of this study will be greatly improved if the analysis of sex and age as independent variables is taken into consideration in the revised version of the manuscript.

Reply: Thank you for these insights. The sex and age relevance is known to us. But since this retrospective article focusses on the predictive power of hsTnT in general, these factors were not taken into account. This will also be a topic of further (prospective) study. This is also added to the limitations section.

Reviewer 2 Report

The manuscript of de Jongh et al. entitled "The Predictive Power of the 14-51 ng/L High Sensitive Troponin T (hsTnT) Values for Predicting Cardiac Revascularization in a Clinical Setting" has the aim to investigate the role of high-sensitivity cardiac troponin T in the observe zone.

The manuscript is well written, however there are several important issues to address.

Major comments:

1. Patient population

- What percentage of patients were excluded due to loss of follow up or missing laboratory parameters. A high percentage of these would be a relevant source of selection bias, please comment this in the limitations section.

- Who decided whether complaints were not of cardiac origin? Were these the treating physicians knowing all the troponin values? This would be a serious limitation introducing a substantial bias and has to be addressed in the limitations section.

2. Baseline parameters

- In Table 1, there seems to be incoherence in numbers and %, please clarify (e.g. 69 CABG patients represent 19 % of 75?)

3. Outcome variable

- How many patients received Echo, stress test (Stress-Echo, MRI, nuclear imaging) or CT-CA? How many of CABG patients received invasive CA and how many CT-CA?

4. Laboratory parameters

- Please clarify how troponin was measured: specimen (serum, plasma, EDTA, etc.), analytics (assay type, assay model/manufacturer, analytical validation of these, e.g. conventional vs. high-sensitivity hs-cTn).

Author Response

Reviewer 2.

  1. Patient population

- What percentage of patients were excluded due to loss of follow up or missing laboratory parameters. A high percentage of these would be a relevant source of selection bias, please comment this in the limitations section.

Reply: This is added to the limitations section.

- Who decided whether complaints were not of cardiac origin? Were these the treating physicians knowing all the troponin values? This would be a serious limitation introducing a substantial bias and has to be addressed in the limitations section.

Reply: Patients are admitted with specific complaints, e.g. dyspnea or chest pain. Since blood is drawn upon entry, no physician is present to order blood test, all patients who are present with these complaints are subjected to these tests. ED physicians and cardiologists have clearly stated through anamnesis in the medical file that these patients were not suspect for cardiac pathology. This is added in the text.

  1. Baseline parameters

- In Table 1, there seems to be incoherence in numbers and %, please clarify (e.g. 69 CABG patients represent 19 % of 75?)

Reply: Percentages have been corrected.

  1. Outcome variable

- How many patients received Echo, stress test (Stress-Echo, MRI, nuclear imaging) or CT-CA? How many of CABG patients received invasive CA and how many CT-CA?

Reply: This data was not available to us and cannot be recovered due to the time that data was gathered

  1. Laboratory parameters

- Please clarify how troponin was measured: specimen (serum, plasma, EDTA, etc.), analytics (assay type, assay model/manufacturer, analytical validation of these, e.g. conventional vs. high-sensitivity hs-cTn).

Reply: available data is added to the Methods section

Round 2

Reviewer 1 Report

To the Authors

Revised manuscript

Unfortunately, Authors revised the manuscript by following only in part my suggestions. In particular, I suggested to add to the revised version of the manuscript that: “… Some indications on the analytical characteristics and performance of the hs-cTnT method used in this study should be reported in details, in particular: the limit of detection (LoD), the 99thpercentile value, and the analytical error in the range from 7 ng/L to 51 ng/L”. These specific data about the analytical performance of assay method used for cTnT assay is fundamental to understand the pathophysiological interpretation associated to the cut-off of 7 ng/L for cTnT chosen by Authors in this study.  Authors state that the cTnT concentrations were measured in the period 2012 and 2013 with a hs-cTnT method. The term “hs-cTnT” should be used only for high-sensitivity cTnI (hs-cTnI) or cTnT (hs-cTnT) methods according to the international AAC and IFCC guidelines (Wu AHB et al. Clinical laboratory practice recommendations for the use of cardiac troponin in acute coronary syndrome: Expert opinion from the Academy of the American Association for Clinical Chemistry and the Task Force on Clinical Applications of Cardiac Bio-Markers of the International Federation of Clinical Chemistry and Laboratory Medicine. Clin Chem 2018;64:645-55). The last generation of cTnT methods with high-sensitivity characteristics was released by Roche Diagnostic only after 2017 (the FDA approved the method named Elecsys Troponin T Gen 5 STAT, with high-sensitivity characteristics for cobas system analyzers only in the January 31, 2017).  This hs-cTnT method has a LoD value of 3 ng/L as reported by the manufacturer (REF 08469873190 for Elecsys Troponin T hs for cobas e801, date 2019-06) and confirmed in an authoritative review using standardized experimental protocols (Clerico A et al. Evaluation of analytical performance of immunoassay methods for cardiac troponin I and T: From theory to practice. Adv Clin Chem 2019;93:239-62)Authors must clearly state the method used for cTnT assay. If the method used is not a high-sensitivity method (as I should presume if the measurement of biomarker was performed in 2012-2013) the cut-off value of 7 ng/L chosen by Authors in this study is very close of the LoD value of the cTnT method used. Accordingly, the > 90% of healthy women and >50% of healthy mean show non-measurable cTnT values at the cut-off of 7ng/L chosen by the Authors in this study (Clerico A et al. The 99th percentile of reference population for cTnI and cTnT assay: methodology, pathophysiology and clinical implications. Clin Chem Lab Med 2017;55:1634-1651). In conclusion, Author should clearly state the revised version of the manuscript the cTnT method used in this study the limit of detection (LOD), the 99th percentile value, and the analytical error in the range from 7 ng/l to 51 ng/L. value of this method evaluated in the reference laboratory of the study or reported by the manufacturer or data from literature.

Furthermore, Authors should report the reference of the 2020 NICE guidelines (NICE. High-sensitivity troponin tests for the early rule out of NSTEMI. Diagnostics guidance. Published: 26 August 2020 www.nice.org.uk/guidance/ dg40). 

Author Response

Dear Reviewer, we’ve added the limits of detection to the Methods section > “Sample collection and Laboratory testing” Roche Diagnostics electrochemiluminescence immunoassays was used. Furthermore the cut off we used in our study for intermediary hsTnT was 14ng/L with the lower limit of detection being <5ng/L. hsTnT values of 5ng/L <> 14ng/L were deemed ‘lower’ according to the in-house protocol and were therefore not included in our study. Our study only sought to analyse the predictive value of hsTnT values between 14ng/L and 51ng/L, where it is unclear if an ACS is present. Therefore we hope that this answer will provide clarity to your question.

Lastly, we have added the 2020 NICE guidelines to the discussion